# DÉJÀ VU: AN EMPIRICAL EVALUATION OF THE MEMORIZATION PROPERTIES OF CONVNETS

## ABSTRACT

Convolutional neural networks memorize part of their training data, which is why strategies such as data augmentation and drop-out are employed to mitigate over-fitting. This paper considers the related question of "membership inference", where the goal is to determine if an image was used during training. We consider membership tests over either ensembles of samples or individual samples.

First, we show how to detect if a dataset was used to train a model, and in particular whether some validation images were used at train time. Then, we introduce a new approach to infer membership when a few of the top layers are not available or have been fine-tuned, and show that lower layers still carry information about the training samples. To support our findings, we conduct large-scale experiments on Imagenet and subsets of YFCC-100M with modern architectures such as VGG and Resnet.

## 1 INTRODUCTION

The widespread adoption of convolutional neural networks (LeCun et al., 1990) (ConvNets) for most recognition tasks, was triggered by the work of Krizhevsky et al. (2012) in image classification and subsequent deep architectures (Simonyan & Zisserman, 2014; He et al., 2016). Several works have analyzed these architectures from different perspectives. Zeiler & Fergus (2014) have proposed DeconvNet to vizualize filter activations. Lenc & Vedaldi (2015) analyze their equivariance. Mahendran & Vedaldi (2015) show how to invert them and synthetize images maximizing the response of different classes. Ulyanov et al. (2018) analyze the image priors implicitly defined by ConvNets.

All these works increase our understanding of ConvNets, but the complex issue of overfitting and its relationship to optimization are still not fully understood. Several strategies are routinely used to avoid overfitting, such as $\ell_2$-regularization through weight decay (Krogh & Hertz, 1991), dropout (Srivastava et al., 2014), and importantly, data augmentation (Behpour et al., 2017; Dwibedi et al., 2017; Paulin et al., 2014). Yet few works (Zhang et al., 2017; Yeom et al., 2018) have analyzed the interplay of overfitting and memorization of training images in high-capacity classification architectures. Specifically, we are not aware of such an analysis for a modern ConvNet such as ResNet-101 learned on Imagenet.

In this paper, we consider the privacy issue of membership inference, *i.e.*, we aim at determining if a specific image or group of images was used to train a model. This question is important to protect both the privacy and intellectual property associated with images. For ConvNets, the privacy issue was recently considered by Yeom et al. (2018) for the small MNIST and CIFAR datasets. The authors evidence the close relationship between overfitting and privacy of training images. This is reminiscent of prior membership inference attacks, which employ the output of the classifier associated with a particular example to determine whether it was used during training or not (Shokri et al., 2017). This is related to Torralba & Efros (2011), who showed that a classifier can determine with high accuracy if an image comes from a dataset or another by exploiting the bias inherent to datasets. We discuss this relationship and show that we can detect whether a given network has been trained on some of the validation images. This has a concrete application for machine-learning benchmarks: scores are often reported on a validation set with public labels, allowing a malicious or gawky competitor to artificially inflate the accuracy by training on validation images. Our test detects if it is the case, even if only part of the validation set is leaked to the training set.

Finally, we propose a new setting for membership inference that only considers intermediary layers of a network, thus extending membership inference to transferred and fine-tuned networks, that have become ubiquitous. Our membership inference does not require the last layer(s) of the original ConvNet to perform the test. This is important because, in many contexts, image recognition systems are built upon a trunk trained on a dataset and then fine-tuned for another task. Examples include Mask-RCNN (He et al., 2017) and models used for fine-grained recognition (Hariharan & Girshick, 2017). In both cases there are not enough training samples to train a full network: only the last layers of the networks are fine-tuned. In summary, our paper makes the following contributions:

- A simple statistical test to detect the "signature" of a dataset in a trained convnet, and to detect if validation images where used to train the model (leakage).
- A membership inference test that detects if an image was used to train the trunk of a network. To our knowledge, it is the first work on membership inference that attacks intermediate layers.

The paper is organized as follows. Section 2 reviews related work. Section 3 formally introduces the problem. Section 4 considers the problem of determining if a particular dataset, *e.g.*, the validation set, was used during training. Section 5 focuses on detecting if a particular image has been used for training without accessing the network's output layer.

## 2   RELATED WORK & DATASETS

Our work is related to the topics of overfitting and memorization capabilities of neural network architectures, which are able to perfectly discriminate random outputs in some cases (MacKay, 2002; Zhang et al., 2017). In the following, we distinguish *explicit* from *implicit* memorization (also called "unintended memorization" (Carlini et al., 2018) in natural language processing systems).

**Explicit memorization.**   Neural network are capable of memorizing any random pattern. This property was analyzed in MacKay (2002) for the single layer case. In MacKay's setup, the sender and receiver agree beforehand on a set of vectors $(x_i)_{i=1}^n \in \mathbb{R}^d$. To transmit an arbitrary sequence of binary labels $y_1, \ldots, y_n$, the sender learns a single-layer neural network that predicts the $y_i$ from $x_i$, and sends its weights to the receiver. The receiver labels the points $x_1, \ldots, x_n$ with the transmitted neural network to reconstruct the labels. The VC-dimension of this 1-layer model is $d$, so the model can fit perfectly as long as $n \leq d$. MacKay extends this bound by showing that the sender can, with high probability, find a neural network fitting the output if $n \leq 2d$, and that it is almost impossible to fit the model for $n > 2d$. The estimated capacity of this neural network is thus 2 bits per parameter.

Determining the practical memorization capacity of ConvNets is not trivial. A few recent works (Zhang et al., 2017; Yeom et al., 2018) evaluate how a network can fit random labels. Zhang et al. (2017) replace true labels by random labels and show that popular ConvNets can perfectly fit them in simple cases, such as small datasets (CIFAR10) or Imagenet without data augmentation. Krueger et al. (2017) extend their analysis and argue in particular that the effective capacity of ConvNets depends on the dataset considered. In a privacy context, Yeom et al. (2018) exploit this memorizing property to watermark networks. As a side note, random labeling and data augmentation have been used for the purpose of training a network without any annotated data (Dosovitskiy et al., 2014; Bojanowski & Joulin, 2017). Our paper is also related to few works (Kraska et al., 2017; Iscen et al., 2017) that learn indexes as an alternative to traditional structures such as Bloom Filters or B-trees. In particular, Kraska et al. (2017) show that in some cases, neural nets outperform cache-optimized B-tree on real-world data. These works exploiting explicit memorization of neural networks are reminiscent of works (Hopfield, 1982; Personnaz et al., 1986; Hinton et al., 1986; Plate, 1995) on associative memories and, more generally, distributed representations.

**Implicit memorization and privacy risk in learning systems.**   Ateniese et al. (2015) state: "*it is unsafe to release trained classifiers since valuable information about the training set can be extracted from them*". The problem that we address in this paper, *i.e.*, to determine whether an image or dataset has been used for training, is related to the privacy implications of machine learning systems. They were discussed in the context of support vector machines (Rubinstein et al., 2009; Biggio et al., 2014). In the context of differential privacy (Dwork et al., 2006), recent works (Wang

et al., 2016; Bassily et al., 2016) suggest that guaranteeing privacy requires learning systems to generalize well, *i.e.*, to not overfit. Note that there are systems providing differential privacy but that still leak information (Ateniese et al., 2015; Balu et al., 2014).

**Membership Inference in images.** A few recent works (Abadi et al., 2016; Hayes et al., 2017; Shokri et al., 2017; Long et al., 2018) have addressed "membership inference" for images: determine whether an image has been used for training or not. Yeom et al. (2018) discuss how privacy, that can be broken by membership inference, is connected to overfitting. Long et al. (2018) observe that some training images are more vulnerable than others and propose a strategy to identify them. Hayes et al. (2017) analyze privacy issues arising in generative models. Most of these works were evaluated on small datasets like CIFAR10, or larger datasets but without data augmentation. Our work aims at being closer to realistic conditions. In the following, the analysis of a pre-trained network will be called "attack" performed by an "attacker".

**Dataset bias and inference.** Torralba & Efros (2011) evidence that a simple classifier can predict with high accuracy which dataset an image comes from. Tommasi et al. (2017) show that this bias still exists with ConvNets. In the next section of this paper, we re-visit this problem by proposing a *dataset inference* method derived from an elementary membership inference test.

**Datasets used in our study.** We use will several public image collections throughout our paper. ***Imnet1k*** refers to the subset of Imagenet (Deng et al., 2009; Russakovsky et al., 2015) used during the ILSVRC-12 challenge. It consists of 1000 balanced classes, split in a training set (1.2M images) and a validation set (50k images). We use the regular split between train and val and denote them by ***Imnet1k-train*** and ***Imnet1k-val***, respectively. ***Imnet22k*** refers to the full Imagenet dataset. It is built in the same way as *Imnet1k*, but with 21783 unbalanced classes. ***Yfcc100M*** (Thomee et al., 2016) contains 99.2M photos that have not been collected for image classification and thus are not representing specific classes or visual concepts. ***Tiny* images** (Torralba et al., 2008) consists of 79M low-resolution images. ***CIFAR10*** is a subset of *Tiny* that has been labelled for image classification. In our study, it is important that the dataset does not contain duplicate images or images that overlap between the train and the test set. We have sanitized the datasets to avoid this problem using GIST descriptors and similarity search, see Appendix B in the supplemental material for details.

## 3 MODEL

In this paper, we consider a set of samples $(z_1, \ldots, z_n)$ along with their set membership information, which consists of binary labels $(m_1, \ldots, m_n)$. Our goal is to assess set membership $(m'_1, \ldots, m'_k)$ for a set of $k$ fresh samples $(z'_1, \ldots, z'_k)$. A sample $z$ is either an image $x$ or an image-label pair $(x, y)$. Note that we do not necessarily have samples $z_i$ for which $m_i = 0$, *i.e.*, we include the case where only positive samples are given.

We consider two cases: in the first case, we make the assumption that all elements $z'_i$ belong to the same set, and thus $m'_1 = \cdots = m'_k$. In the second case, there is no relation between the set membership of different elements, and thus we infer set membership $m'_j$ independently for each $z'_j$. Membership inference methods for groups of images (the first case) are presented in Section 4 for the case where we do not have labels $y$, and we both setups where we have positive and negative samples, or only positives. Section 5 considers the case of individual images (second case), for which we dispose of both the image and the label.

### 3.1 PARAMETRIC MODELS FOR MEMBERSHIP INFERENCE

We describe here the general framework of the membership inference methods used in Section 5, and make the connection with explicit memorization. We assume that we are given a parametrized function $f_\theta$ which extracts features from images. It was typically trained as part of a classifier, such as a neural network. Given image-label pairs $((x_1, y_1), \ldots, (x_n, y_n))$ and their binary set membership $(m_1, \ldots, m_n)$, we aim at learning a model $d_\Lambda$ that predicts membership $m_i$ from features of the images $f_\theta(x_i)$ and the label $y_i$. To optimize $d_\Lambda$ we solve the following optimization problem:

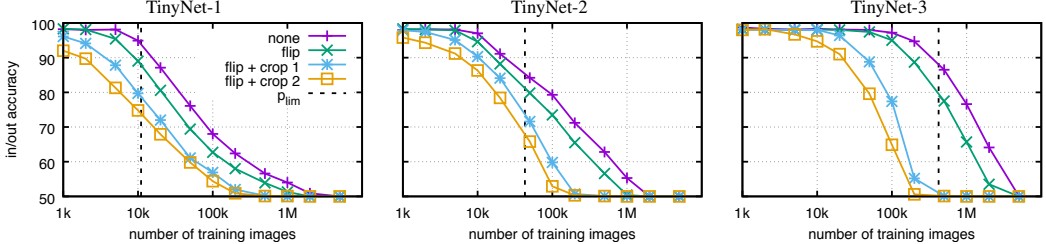

Figure 1: In/out classification performance (train) on *Tiny*, for varying image subsets and number of images. The colors indicate the type of data augmentation: purple=none, green=flip, cyan=flip+crop$\pm$1, orange=flip+crop$\pm$2. The vertical line shows the number of positive images $p_{\text{lim}}$ such that $C(p_{\text{lim}})$ is the number of parameters of the network.

$$\min_{\Lambda} \frac{1}{n} \sum_{i=1}^{n} \ell(d_{\Lambda}(f_{\theta}(x_i), y_i), m_i), \tag{1}$$

where $\ell$ is the cross-entropy loss. The solution of this optimization relies on the difference between the distribution of features from training and that of the held-out set. This task is difficult in general, because set membership is decided independently of the content of the image, and thus $m_i$ is independent from $(x_i, y_i)$. However, the set membership $m_i$ is not independent of the features $f_{\theta}(x_i)$, because $m_i$ has an influence on the parameters of the model $\theta$.

### 3.2 EXPLICIT MEMORIZATION

If we omit the dependency in $y_i$ in the optimization problem 1, the problem relies on separating images through their features $f_{\theta}(x_i)$. Thus a lower bound of this problem is:

$$\min_{\theta} \min_{\Lambda} \frac{1}{n} \sum_{i=1}^{n} \ell(d_{\Lambda}(f_{\theta}(x_i)), m_i). \tag{2}$$

This corresponds to the case of *explicit memorization*: the model $d_{\Lambda}(f_{\theta}(\cdot))$ is trained to separate an arbitrary set of images from the others: it approximates the set membership function and predicts membership $m$ from image $x$. Note that the what we call memorization in this context is different than the memorization discussed in Zhang et al. (2017): we explicitly approximate the set membership function, whereas Zhang et al. (2017) fit random labels to the data. This optimization problem bears similarity with GANs, with two key differences: GANs perform a mini-max optimization, and the discriminator is trained to separate generated samples from "true" samples from the distribution.

If we denote by $p$ the number of positive samples (for which $m_i = 1$) and $N$ the total number of images available, the quantity of information contained in such a set is:

$$C(p) = \log_2 \binom{N}{p} \approx p \log_2 \left(\frac{N}{p}\right) + \frac{p}{\log(2)}, \tag{3}$$

where the approximation holds for $p \ll N$. This allows us to perform experiments by varying the number of positive samples $p$ (and thus the quantity of information $C(p)$) and measuring whether it is possible to fit a model.

We experimented with VGG-type architectures on TinyImages, and VGG-16 and Resnet-101 on Imagenet, see Appendix C for details. Figure 1 shows the accuracy of a trained neural network as a function of the number of samples. The vertical bar shows the number of positive samples $p_{\text{lim}}$ for which the quantity of information $C(p_{\text{lim}})$ is equal to the number of parameters of the network. Observe the impact of data augmentation: models trained on data-augmented images perform less

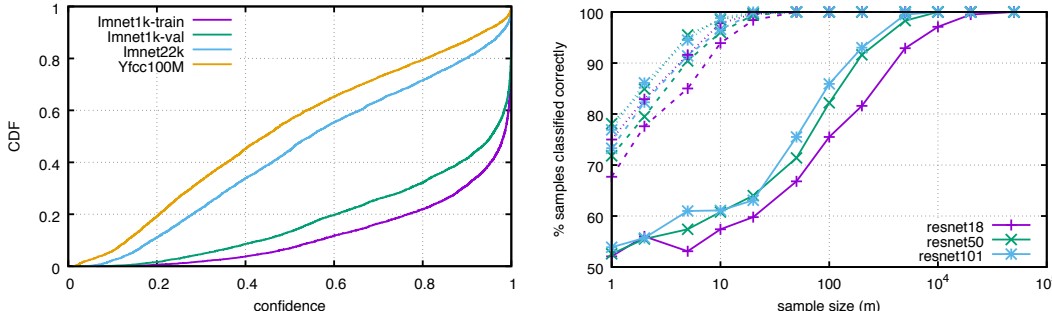

Figure 2: *Left:* Cumulative distribution of the maximum classification score for a sample of 5000 images taken from 4 datasets. *Imnet1k-train* served as the training set and therefore *Imnet1k* images (both train and val) have higher confidence. *Right:* binary classification accuracy (%) of a sample of $m$ elements from the training set *Imnet1k-train* w.r.t. three other datasets: *Imnet1k-val* (solid), *Imnet22k* (dashed) and *Yfcc100M* (lines). The architecture is indicated by the line color.

well than their counterpart trained on plain images. However, fitting a data-augmented model is easier than fitting a correspondingly higher number of samples: for example, it is easier to fit $p$ positives with flips than $2p$ positives. Data augmentation affects membership inference models similarly, see Section 5 for details. Experiments on Imagenet architectures show that such models are able to perfectly fit a large number of samples (up to $n = 10^5$), even with data augmentation.

## 4 DATASET DETECTION AND LEAKAGE

In this section, we detect whether a group of samples or a dataset has been used to train a model. This problem encompasses the particular case of dataset bias (Torralba & Efros, 2011) and is more difficult, as we need to distinguish datasets even if they share the same statistics, acquisition procedure and labelling process. For instance, we want to be able to determine if images from the validation set of *Imnet1k* were used at train time.

We conduct our attacks using the maximal activation of the softmax layer (aka. the *confidence* of the model) as the features $f_\theta$. This quantity is related to the probability of the correct class, especially in the case where the classifier is perfectly accurate, in which the two quantities are equal. It was shown by Yeom et al. (2018) in the context of membership inference of individual images that the probability of the correct class (or the loss, up to a $-\log$ operator) is as good as the output of the model (the whole softmax) in determining if an image was seen by the model.

The cumulative distribution of the confidence for a model trained on *Imnet1k-train* is shown in Figure 2: most samples coming from the source *Imnet1k-train* have a very high confidence, while the distribution of the source *Imnet1k-val* is more balanced and unrelated sources (*Yfcc100M*, *Imnet22k*) tend to have a more uniform distribution.

**Attack scenarios.** We consider two attack scenarios on the model $f_\theta$. In the first scenario, we have known samples with $m \in \{0, 1\}$, and we guess the value of membership $m$ for a set of fresh samples $x'_1, \ldots, x'_k$. We compare the performance of the attack discriminating *Imnet1k-train* and *Imnet1k-val* to easier attacks discriminating more different sources (*Imnet22k*, *Yfcc100M*) in this scenario. In the second scenario, we only have samples for which $m = 1$ (samples from the test set of *Imnet1k*), and we analyze whether elements from the validation set follow the same distribution. If they do not, it means that some of the elements of the validation set have been used to train the model (we refer to this as *leakage*).

We compare confidence distributions using the Kolmogorov-Smirnov (K-S) distance. Given two cumulative distributions $F$ and $G$, the K-S distance is $d_{\text{KS}}(F, G) = \sup_x |F(x) - G(x)|$. We use the K-S distance to determine if two distributions are similar.

Table 1: Kolmogorov-Smirnov tests on *Imnet1k* validation and test sets for various levels of leakage. The K-S test provides a level of significance ($p$-value) rather than a yes/no answer. Lower values indicate high confidence that the validation and test sets distributions are different, hinting that leakage has occurred. If only 1 image per class of the validation set has leaked, we cannot conclude from this test that there has been leakage. Conversely, when 10 images or more have leaked, we can conclude with high significance that leakage has occurred.

| Nb. of Images # per class leaked | Resnet-18 | VGG-16 |
|---|---|---|
| 1 | 0.888 | 0.494 |
| 2 | 0.228 | 0.107 |
| 5 | 0.068 | 0.014 |
| 10 | $< 10^{-4}$ | $< 10^{-4}$ |
| 20 | $< 10^{-4}$ | $< 10^{-4}$ |

### 4.1 CONFIDENCE AS A SIGNATURE OF A DATASET

In this section, we denote by $S_0$ (resp. $S_1$) the set of samples for which $m = 0$ (resp. $m = 1$), and we assume that all samples $\{x'_1, \ldots, x'_k\}$ come from either source $S_0$ or $S_1$. The attacker uses the following decision rule: compute the K-S distance between $x'_1, \ldots, x'_k$ and $S_0$ (resp. $S_1$), and assign the samples to the closest source.

**Results and observations.** The results are reported in Figure 2. We can distinguish *Imnet1k-train* from *Yfcc100M* with very few (10-20) samples. More interestingly, the same number of samples allow us to separate *Imnet22k* from *Imnet1k-train*, and with 500 images we can distinguish *Imnet1k-train* from *Imnet1k-val*. This shows that, even with a relatively low number of images, an attacker can determine that a given image collection was used for training. The figure also shows that the test is easier for networks with a higher capacity, that tend to overfit more.

### 4.2 DETECTING LEAKAGE

We now assume that we are given a model for which we suspect that part of the validation set was used for training. For a number of datasets (*e.g.*, Imagenet, Pascal VOC), the labels of the validation set are publicly available, and models are often compared using validation accuracy. A malicious person could train a model using the training set and part of the validation set, and then report validation accuracy to artificially inflate the performance of the model.

The attack we propose is a two-sample K-S test to determine if leakage has occurred or not. We assume that no sample from the test set has leaked (labels are not public in most cases). The null hypothesis of our test is that the validation and test sets have the same distribution. We compute the K-S distance between the validation and test sets, and reject the null hypothesis if this distance is high. The distance threshold $t$ is set such that the null hypothesis is incorrectly rejected with a low probability $\alpha$, corresponding to the $p$-value. For large samples, Smirnov's estimate of the threshold corresponding to a $p$-value of $\alpha$ is (Feller, 1949):

$$t = c(\alpha)\sqrt{\frac{n+m}{nm}} \quad \text{where} \quad c(\alpha) = \sqrt{-\frac{1}{2}\log\left(\frac{\alpha}{2}\right)}. \tag{4}$$

We ran experiments on Imagenet using Resnet-18 and VGG-16, with $s \in \{1, 2, 5, 10, 20\}$ images per class of the validation set in addition to the training set to fit the model. Table 1 reports the $p$-value of the different tests. We can see that when 10 images per class are leaked, the K-S test predicts that leakage has happened with a very high significance. When 5 images per class or less are used, we cannot reject the null hypothesis and thus cannot claim that leakage has happened.

## 5 IMPLICIT MEMORIZATION & MEMBERSHIP INFERENCE

This section tackles the more difficult problem of membership inference in trained models. From a trained model and an image the attacker has to determine whether the image was used to train the model. In our new setting, upper layers are not available (due to *e.g.* finetuning on a downstream task). We provide baselines for VGG16 and Resnet and extend the traditional attacks to our setup.

The literature (Abadi et al., 2017) distinguishes two cases types of membership inference: (1) all layers are available (*all-layers*), (2) only the final output of the network is available (*final-output*). There is currently no attack that performs substantially better in *all-layers* than in *final-output*. This seems counter-intuitive but we confirmed it in preliminary experiments. Our new setup, *partial-layers*, is adapted to transfer learning: only a certain number of bottom layers are available for attack, the remaining layers were destroyed by retraining on an unrelated task. This task is more difficult than *all-layers* since it has less parameters available, and thus more difficult than *final-output*.

### 5.1 EVALUATION PROTOCOL AND BASELINES

We assume that there are three disjoint sources of data: a *public* set, a *private* set, and an *evaluation* set. A model is trained on the private set. The attacker has access to the lower layers of this model and to the public set. After the attack is carried out, the evaluation is run on images from the private and evaluation sets. Note that the public set can be used in a number of different ways by the attacker: in our method, it will be used to retrain the missing layers.

We divide *Imnet1k* equally into two splits (each with half of the images per class). We hold out 50 images per class in the first split to form the evaluation set, and form the private set with the rest of this split. The second split is used as the public dataset. We conduct the membership inference test by comparing the prediction of the attack model on the private set and on the evaluation set. For this purpose, we consider the two baseline methods.

**Bayes rule.** A simplistic membership inference attack is to predict that an image comes from the training set if its class is predicted correctly, and from a held-out set otherwise. We note $p_{\text{train}}$ (resp. $p_{\text{test}}$) the classification accuracy on the training (resp. held-out) set, and assume a balanced prior on membership. According to Bayes' rule, the accuracy of the heuristic is (see Appendix A in the supplementary material for the derivation):

$$p_{\text{bayes}} = 1/2 + (p_{\text{train}} - p_{\text{test}})/2. \tag{5}$$

Since $p_{\text{train}} \geq p_{\text{test}}$ this heuristic is better than random guessing (accuracy $1/2$) and the improvement is proportional to the overfitting gap $p_{\text{train}} - p_{\text{test}}$.

**Maximum Accuracy Threshold (MAT).** Yeom et al. (2018) propose an attack on the loss value: a sample is deemed part of the training set if its loss is below a threshold $\tau$. If $F_{\text{train}}$ (resp. $F_{\text{heldout}}$) is the cdf of the loss on the train (resp. held out), the accuracy of the MAT is:

$$p_{\text{threshold}} = \max_{\tau} 1/2 + 1/2 \left( F_{\text{train}}(\tau) - F_{\text{heldout}}(\tau) \right) \tag{6}$$

As $F_{\text{train}}(\tau) \geq F_{\text{heldout}}(\tau)$, this heuristic is also better than random guessing. In practice, $\tau$ is estimated with samples or simulated by training models with known train/heldout split.

### 5.2 MEMBERSHIP INFERENCE WITH A TRUNCATED NETWORK

In this section, we provide a simple method to attack networks in the *partial-layers* setting. We use the available public data to retrain the missing layers, and apply either the Bayes attack, as if there was no fine-tuning at all. On this retrained layers, we found the MAT attack performed almost the same as the Bayes attack, but the latter is simpler as it does not require to fit a parameter. We found this method to be more accurate than another variant that we designed with shadow models (Shokri et al., 2017), as detailed in the supplemental material (Appendix E).

### 5.3 EXPERIMENTS ON LARGE CONVNETS

**Classification models.** We experiment with the popular VGG-16 (Simonyan & Zisserman, 2014) and Resnet-101 (He et al., 2016) architectures. The private model is learned in 90 epochs, with an

Table 2: Accuracy of membership inference attacks on the softmax layer of the models (*final-output*). Data augmentation reduces the gap between the training accuracy and the held-out accuracy, thus decreasing the accuracy of the Bayes attack and the MAT attack.

| Model | Augmentation | Bayes baseline | MAT |
|-------|--------------|----------------|-----|
| Resnet101 | None | 76.3 | 90.4 |
| | Flip, Crop $\pm 5$ | 69.5 | 77.4 |
| | Flip, Crop | 65.4 | 68.0 |
| VGG16 | None | 77.4 | 90.8 |
| | Flip, Crop $\pm 5$ | 71.3 | 79.5 |
| | Flip, Crop | 63.8 | 64.3 |

Table 3: Accuracy of membership inference attacks on intermediate layers of Resnet-101 and VGG-16 models (*partial-layers*). *Last block* corresponds to the first fully connected layer for VGG-16 and to the 4-th stage of Resnet-101.

| Augmentation | Truncate | Resnet-101 | VGG-16 |
|--------------|----------|------------|--------|
| None | Softmax | 73.4 | 74.8 |
| | Last block | 53.1 | 51.7 |
| Flip, Crop$\pm 5$ | Softmax | 65.7 | 67.3 |
| | Last block | 53.1 | 52.2 |
| Flip, Crop | Softmax | 60.8 | 58.5 |
| | Last block | 52.9 | 53.2 |

initial learning rate of $0.01$, divided by 10 every 30 epochs. Parameter optimization is conducted with stochastic gradient descent with a momentum of $0.9$, a weight decay of $10^{-4}$, and a batch size of $256$. To assess the effect of data augmentation, we train different networks with varying data augmentation: flip+crop$\pm 5$, flip+crop, flip+crop+resize, or none.

**Attack models.** We evaluate both the Bayes and MAT methods to estimate the performance on *final-output*. The results are shown in Table 2. As we can see, it is possible to guess with a very high accuracy if a given image was used to train a model when there is no data augmentation. Stronger data augmentation reduces the accuracy of the attacks, that still remain above $64\%$.

The results of our attack in the more challenging *partial-layers* setting are shown in Table 3. We can see that even without the last layers, it is possible to infer training set membership of an image. The attack performance depends on two factors: the layer at which the attack is conducted, and the data augmentation used to train the original network. As expected, it is more difficult to attack a network that has been trained with more data augmentation, or that has only lower layers available. More importantly, these experiments show that intermediate layers still carry out information about the images used for training the model.

## 6 CONCLUSION

We have investigated the memorization capabilities of neural networks from different perspectives. Our experiments show that state-of-the-art networks can remember a large number of images and distinguish them from unseen images. We have analyzed networks specifically trained to remember a set of images and the factors influencing their memorizing and convergence capabilities. It is possible to determine whether an image set was used at training time, even with full data augmentation. On the contrary, the accuracy of determining if a single image was used is low when considering full data augmentation on a large training set such as Imagenet. This implies that data augmentation is an effective privacy-preserving method. Our last contribution is a method that detects training images better than chance even with no access to the last layers, under limited data augmentation.

**Final remark:** The curious reader may have noticed that our title echoes the one of a previous user study (Dhamija et al., 2000), in which the authors discussed the feasibility of authenticating humans by their capabilities to recognize a set of images.

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

## APPENDIX A  PROBABILISTIC DERIVATIONS

### A.1  BAYES ATTACK

Let $C$ denote the event that the prediction of the neural network is correct and $S$ the random variable that indicates whether the sample comes from the training set. We therefore have:

$$\mathbb{P}(C = 1|S = 1) = p_{\text{train}}, \mathbb{P}(C = 1|S = 0) = p_{\text{test}} \tag{7}$$
$$\mathbb{P}(S = 1) = \mathbb{P}(S = 0) = 1/2. \tag{8}$$

The accuracy of Bayes attack is:

$$\mathbb{P}(C = S) = \mathbb{P}(C = 1 \mid S = 1)\mathbb{P}(S = 1) \tag{9}$$
$$+ \mathbb{P}(C = 0 \mid S = 0)\mathbb{P}(S = 0) \tag{10}$$
$$= \frac{1}{2}(p_{\text{train}} + 1 - p_{\text{test}}). \tag{11}$$

### A.2  EQUIVALENCE BETWEEN KOLMOGOROV-SMIRNOV AND THRESHOLD ATTACKS

If we consider the particular case of a subset of $m = 1$ image, we show in this section that the decision boundary induced by the K-S distance is the same as the MAT described in Section 5.1. Yet there are two significant differences between the K-S attack and the MAT: we consider *confidence* instead of the loss value, and the optimal threshold is computed differently. Our attacks with the K-S distance can therefore be seen as a generalization of the membership inference proposed by Yeom et al. (2018).

We assume that we have two cumulative distributions $F$ and $G$ such that $\forall x, F(x) \geq G(x)$. We want to show that the K-S rule is equivalent to a threshold rule. Denoting by $\delta_x$ the Dirac distribution centered on $x$, we have:

$$d_{\text{KS}}(\delta_x, F) \leq d_{\text{KS}}(\delta_x, G) \tag{12}$$
$$\iff \frac{1}{2} - |F(x) - \frac{1}{2}| \leq \frac{1}{2} - |G(x) - \frac{1}{2}| \tag{13}$$
$$\iff |G(x) - \frac{1}{2}| \leq |F(x) - \frac{1}{2}|. \tag{14}$$

The two following cases are easy:

$$G(x) \leq F(x) \leq 1/2 \Rightarrow d_{\text{KS}}(\delta_x, F) \leq d_{\text{KS}}(\delta_x, G), \tag{15}$$
$$F(x) \geq G(x) \geq 1/2 \Rightarrow d_{\text{KS}}(\delta_x, F) \geq d_{\text{KS}}(\delta_x, G). \tag{16}$$

For the last case, the set $I$ for which $G(x) \leq 1/2 \leq F(x)$ is an interval. On this interval, $|F(x) - 1/2| - |G(x) - 1/2| = F(x) + G(x) - 1$. $F + G$ is increasing, and thus there exists a threshold $\tau$ such that for $x \in I$:

$$x \leq \tau \iff d_{\text{KS}}(\delta_x, F) \leq d_{\text{KS}}(\delta_x, G). \tag{17}$$

With Equations 15 and 16, Equation 17 extends to all $x$.

## APPENDIX B  DE-DUPLICATING THE DATASETS

In this section, we describe the de-duplication processing applied to the datasets used in explicit memorization experiments. This process ensures that near-duplicate images do not get assigned different labels, and thus makes learning and evaluation more reliable.

### B.1  DESCRIPTION AND MATCHING OF DUPLICATES

We compare images using GIST (Oliva & Torralba, 2006), a simple hand-crafted descriptor that was shown to perform well on moderate image transformations (Douze et al., 2009). We compute the

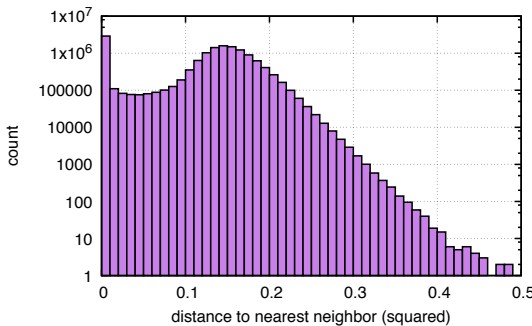

Figure 3: Histogram of distances of the images of *Imnet22k* to their nearest neighbor.

TinyNet 1                    TinyNet 2                    TinyNet 3

Figure 4: Tiny nets.

approximate k-nearest neighbor graph on each dataset using Faiss (Johnson et al., 2017). Figure 3 shows the histogram of distances for the images of *Imnet22k* to their nearest neighbor: the bin around $[0, 10^{-2}]$ contains more images than the following bin $[10^{-2}, 2.10^{-2}]$, which is due to duplicates in the dataset.

Images that are bit-wise exact are unambiguous duplicates – in fact they are often already removed beforehand from the datasets because they are easy to detect by computing a hash value on the content. Beyond this extreme case, the notion of "duplicate" is ambiguous: images that are re-encoded, resized, slightly cropped should be considered duplicates, but the case of larger transformations is less obvious (e.g., photos of the same painting, consecutive frames of a video).

## B.2 IDENTIFICATION OF CONNECTED COMPONENTS

We set a conservative arbitrary threshold of $0.001$ to detect duplicate images, and remove the edges of the k-nn graph that are above this threshold. We compute the connected components, and keep a single image per connected component.

For *Imnet22k*, the largest connected components are error images returned by image banks like Flickr for missing entries. This is an artifact of how the dataset was downloaded. The largest non-trivial cluster from *Imnet22k* is the image of a flower in Figure 5, that appears in 72 different synsets. There seems to be some disagreement on the species of this flower, along with plain bad annotations.

## B.3 STATISTICS

Table 4 shows some statistics on the duplicates identified by our simple approach. *Imnet22k* has 10.4 % duplicate images. In addition to these duplicates, we removed 930,757 images that overlap with *Imnet1k*, which means that *Imnet1k* is not a subset of *Imnet22k* in this paper. Within *Imnet1k*, we found 1 % duplicates, which seems small enough not to remove them. For *Tiny*, we found 9.5 % duplicates and removed them, leaving the dataset with $71, 726, 550$ unique images.

Table 4: Duplicate statistics for the datasets we use.

| Dataset | # images | # groups |
|---|---|---|
| *Imnet22k* | 14,197,087 | 12,720,164 |
| *Imnet1k-train* | 1,281,167 | 1,267,936 |
| *Tiny* | 79,302,017 | 71,726,550 |

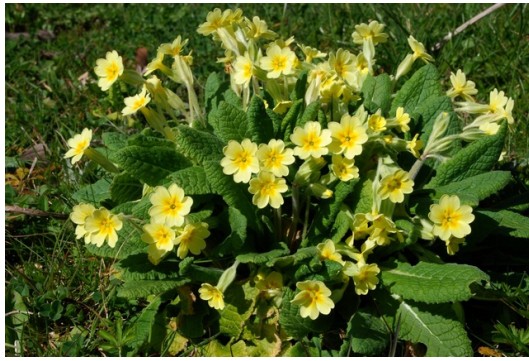

| | |
|---|---|
| n11610437 | bishop pine, bishop's pine, Pinus muricata |
| n11619455 | western larch, western tamarack, Oregon larch, Larix occidentalis |
| n11621281 | amabilis fir, white fir, Pacific silver fir, red silver fir, Christmas tree, Abies amabilis |
| n11626826 | red spruce, eastern spruce, yellow spruce, Picea rubens |
| n11710827 | cucumber tree, Magnolia acuminata |
| n11721642 | lesser spearwort, Ranunculus flammula |
| n11722342 | western buttercup, Ranunculus occidentalis |
| n11722621 | cursed crowfoot, celery-leaved buttercup, Ranunculus sceleratus |
| n11753562 | buffalo clover, Trifolium reflexum, Trifolium stoloniferum |
| n11840476 | desert four o'clock, Colorado four o'clock, maravilla, Mirabilis multiflora |
| n11874081 | yellow rocket, rockcress, rocket cress, Barbarea vulgaris, Sisymbrium barbarea |
| n11882426 | crinkleroot, crinkle-root, crinkle root, pepper root, toothwort, Cardamine diphylla, Dentaria diphylla |
| n11887750 | western wall flower, Erysimum asperum, Cheiranthus asperus, Erysimum arkansanum |
| n11889205 | tansy-leaved rocket, Hugueninia tanacetifolia, Sisymbrium tanacetifolia |
| ... | ... |

Figure 5: Image that appears in the largest number of duplicate versions in *Imnet22k* (72), with a few of the corresponding synsets.

## APPENDIX C    TINYNET ARCHITECTURES

In this section, we explicitly train neural networks to memorize a given subset of images, so that it can decide whether an image is in its memory or not at test time. We design a model $d_\Lambda(d_\theta(\cdot))$ that distinguishes a set of *in* images from *out* images, where images unseen during training are *out*.

We repurpose the classification layer of standard models to output a binary label, depending on whether the image must be remembered or not. Our architecture plays an equivalent role to the discriminator in Generative Adversarial Networks (GAN): it needs to discriminate between positive and negative images. In our case, negative images are a large pool of images instead of the generated images in GANs. Zhang et al. (2017) show that ConvNets are able to overfit almost any random labelling of their input data, but in their experiment, the output for unseen images is undefined.

### C.1    EMPIRICAL ANALYSIS ON TINY IMAGES

**TinyNet.**    We design a family of ConvNets with 4 convolutional layers and 2 fully-connected layers that take 32x32 images as input and output a binary classification. There are 3 versions: TinyNet-1, (90k parameters), TinyNet-2 (300k parameters) and TinyNet-3 (2M parameters). Most parameters of these models are in the first fully connected layer, as in VGG (cf. Appendix C).

**Experimental setup.**    We use a subset of $N = 15M$ images from *Tiny* for these experiments. We randomly sample $n$ images as positive examples, and treat the rest as negatives. At each epoch, we feed a random sample of negatives of the same size as the number of positives to the network. The reported accuracy is measured on a balanced set of positives and negatives. We consider four types of data augmentation: "none", "flip" (random horizontal mirroring), "flip+crop±1" (a random translation in $\{-1, 0, +1\}^2$), "flip+crop±2".

**Discussion.**    Figure 1 shows the accuracy of the model as a function of the number of positive images for all TinyNets. Instead of a sharp drop between the over-capacity and the under-capacity regimes, we observe a smooth drop as the number of positives increases. Empirically, this transition phase happens when the number of samples reaches the theoretical capacity of the network.

As expected, data augmentation reduces the memorization capacity of the network. For example, the accuracy of a network trained on $n$ images with flips is lower-bounded by the capacity of the same network trained on $2n$ images with no data augmentation. This lower bound is not tight, thanks to the generalization capability of the ConvNet, which captures the patterns common to an image and its symmetric. This generalization capability is obvious for stronger augmentations: for example with "flip+crop±1" TinyNet-2 can identify 10k images with 90% accuracy, vs. 20k images without data augmentation, while this requires to treat 18 augmented versions of each image similarly.

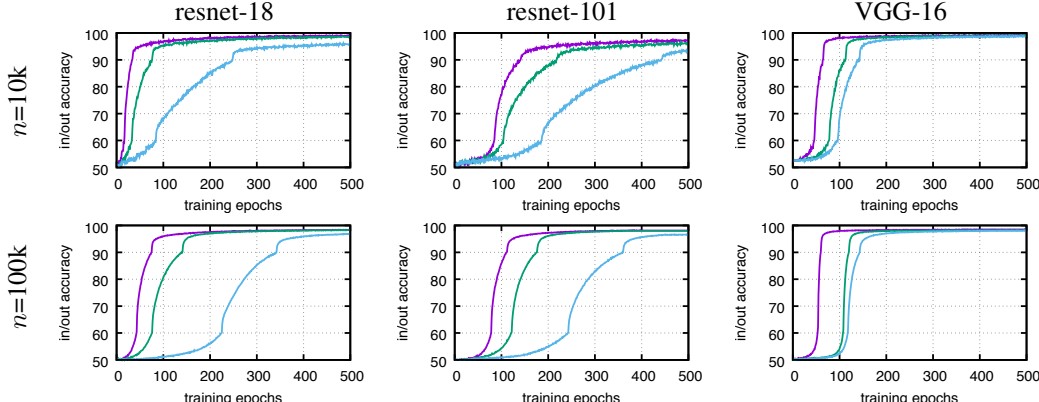

Figure 6: Accuracy over iterations of the in/out training on *Yfcc100M* for different networks and amount of data augmentation (indicated by color: purple= none, green = flip, cyan = flip+crop±5).

## C.2 EXPERIMENTS WITH LARGE-SCALE ARCHITECTURES

In this section, we extend the explicit memorization experiments to VGG-16, ResNet-18, and ResNet-101 networks with images coming from *Yfcc100M*. The capacity of these networks is much larger than in the tiny setting: Resnet-18 has 11.7M parameters and VGG-16 has 140M.

We set an initial learning rate of $10^{-2}$ and divide it by 10 when the training accuracy gets over 60%, and again at 90%. We run experiments using either the center crop, or two data augmentations (flip, flip+crop±5). Figure 6 shows convergence plots for several settings. Note, the x-axis is in epochs, that are $10\times$ slower for $n$ =100k images than $n$ =10k images. The longest experiment took 4 days on 4 GPUs . VGG-16 and ResNet-101 converge at approximately the same number of epochs, irrespective of $n$. Data augmentation increases the number of epochs required to converge, eg. for the ResNets, flip up to twice more epochs to be trained. VGG is a shallower and higher capacity network; in general it converges faster and it handles crops better than the ResNet variants.

The outcome of our analysis is that explicit memorization of a large amount of images is possible, albeit more difficult with data augmentation. In real use cases, the number of images that can be stored explicitly with perfect accuracy is practically much lower than the number of network parameters. This set of experiments provides an approximate upper-bound for the problem of membership inference: if a given model cannot perfectly remember a set of images when trained to do so, it will likely not be able to remember all the images of the training set when trained for classification.

The architectures includes from 3 convolutional layers for TinyNet1 to 4 for TinyNet2 and TinyNet3. The first convolutional layer is 5x5. Each convolutional layer is followed by a Rectifier Linear Unit activation. The fully connected layer of TinyNet3 is larger than TinyNet2.

## APPENDIX D    FILTERS

The filters of the first convolutional layer are easy to visualize and contain interesting information on how the SGD optimized to the very first filter that is applied on the image pixels (Krizhevsky et al., 2012; Bojanowski & Joulin, 2017; Paulin et al., 2017). Figure 7 shows the filters obtained after training a Resnet-18. The filters for 10k images are very noisy compared to the smooth Gabor filters produced by supervised classifiers. This is probably due to the large capacity of the network, that is able to quickly overfit the data and does not need to update the filter weights beyond their random initialization. With more images, the filters become more uniform, exhibiting some specialization. Interestingly, for $n$=100k with crop augmentation the filters have a clear uniform color. This is required for the output to be less sensitive to translations of up to 2 pixels.

$n$=100k, no augmentation   $n$=100k, flip   $n$=100k, flip+crop$\pm2$

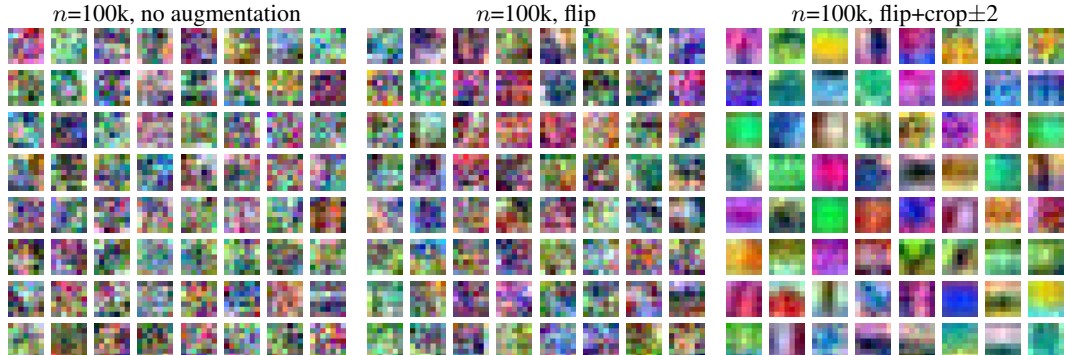

Figure 7: Filters of the first convolutional layer (7x7, 64 filters) obtained when learning to explicitly memorize if an image was used for training or not.

Table 5: Accuracy of membership inference attacks before the softmax layer of the models (*partial-layers*), using shadow models.

| Model | Augmentation | Attack accuracy |
|---|---|---|
| Resnet101 | None | 60.6 |
| | Flip, Crop $\pm5$ | 61.4 |
| | Flip, Crop | 58.2 |
| VGG16 | None | 73.8 |
| | Flip, Crop $\pm5$ | 65.8 |
| | Flip, Crop | 55.2 |

## APPENDIX E    SHADOW MODELS

We evaluated the performance of shadow models on the partial-layers setting. The setting is the following: we train 20 networks on the public dataset, each time holding out a different subset of images. For each network, we can thus compare the activations of train and held-out images. These activations are not directly comparable between two different networks, because internal activations of a ReLU network have invariances (such as permutation of the neurons or positive rescaling). To circumvent this issue, we learn a regression model that maps activations between two networks, and thus align activations of all the networks to the activations of the network under attack using the $\ell_2$ loss. We then learn an attack model that predicts from the aligned activations whether the image was seen by the network at train time.

The results are shown in Table 5. While performing better than random guessing, shadow models underperform the attack methods shown in Table 3. We believe that this is due to the complex processing involved in training shadow models on intermediate activations (notably the regression model), whereas the attacks of Section 5 are more straightforward to train.

