# OpenReview forum: "Déjà Vu: An Empirical Evaluation of the Memorization Properties of Convnets"
_ICLR.cc/2019/Conference_

### Official Review · AnonReviewer1 · 2018-10-24
**Review of "an empirical evaluation of the memorization properties of convnets"**

**Rating:** 6
**Confidence:** 2

**Review:**

I read the other reviewers' comments as well as the rebuttal. I think that the other reviewers make a number of valid points, especially with regards to the theoretical analysis of the paper. Therefore, I do not feel confident in championing this paper.

PS: I am downgrading my confidence in my evaluation.

---

Paper 93 proposes an empirical evaluation of the memorization properties of convnets. More specifically, it evaluates three aspects:
-	First it evaluates whether convnets can learn to distinguish images from two different sets by training a binary classifier. The conclusion is that, indeed, deep convnets can learn to make such a decision. As could be guessed from intuition, the larger the capacity of the network and the smaller the size of the sets, the higher the accuracy.
-	Second, it evaluates whether we can detect that a group of samples of a dataset was used to train a model. For this purpose, it is proposed to compute the distribution of maximal activation scores of the output softmax layer and to make use of the Kolmogorov-Smirov distance between the cumulative distributions. It is shown experimentally that one can detect (even partial) leakage with such a technique.
-	Third, it evaluates whether we can detect that a single images was used to train a convnet. Two simple techniques are proposed. The first one considers that a sample is part of the training set if it correctly classified. The second one considers that a sample is part of the training set if its loss is below a threshold. It is shown experimentally that one can make such a decision with moderate accuracy.

On the positive side:
-	This is a topic that should be of broad interest to the ICLR community.
-	The paper is generally well-written.
-	The experiments are reported on large-scale datasets on high-capacity networks which is more realistic than small-scale settings.

On the negative side:
-	It is unclear whether the data augmentation techniques is applied only at training time or also at test time. In other words: at test time, do you present the original images only or transformed images too?
-	In section 4, it is unclear why only the maximal activation of the softmax layer is used to characterize a sample? Why not considering the full distribution that should contain richer information? Why just focusing on the output layer and why not using the info available at intermediate layers?
-	Section 5 is somewhat less clear than the previous sections. The authors should more clearly define what the private, public and evaluation sets are, right from the beginning. The purpose of the public set is explained only in section 5.2.
-	The experimental results of section 5.2 are somewhat disappointing. Even with no data augmentation, and even with the original networks, membership can only be assessed with a 90% accuracy. Results are much lower in less favorable cases, sometimes close to random (see last line of Table 3). This seems to be too low to be of practical use. This might be because the Bayes and MAT attacks are too simplistic. Again, why not using the distribution of the outputs of all layers? Why focusing only on the output of the last layer?

---

> ### Author Response · Authors · 2018-11-20
> **Answer to reviewer 1**
>
> We thank the reviewer for their review. We address the different remarks below.
>
> “It is unclear whether the data augmentation techniques is applied only at training time or also at test time. In other words: at test time, do you present the original images only or transformed images too?”
> We apply the data augmentation both at training and test time.
>
> “In section 4, it is unclear why only the maximal activation of the softmax layer is used to characterize a sample? Why not considering the full distribution that should contain richer information? Why just focusing on the output layer and why not using the info available at intermediate layers?”
> We agree that the full distribution of the softmax layer provides more information, but there is no straightforward way to extend the Kolmogorov-Smirnov distance to multi-dimensional distributions, beyond the two- and three-dimensional cases. We focus on confidence as a proxy to the loss, and we assume that the loss is the quantity that should be the most different between training and testing, as the optimization phase explicitly minimizes the loss on the training set. Moreover, early experiments showed that using the outputs of intermediate layers provide no improvement for membership inference (on preliminary CIFAR-10 experiments, we obtained respectively 67.7 accuracy with the output layer and 66.5 when using all layers).
>
> “Section 5 is somewhat less clear than the previous sections. The authors should more clearly define what the private, public and evaluation sets are, right from the beginning. The purpose of the public set is explained only in section 5.2.”
> We will update this section to make it clearer.
>
> “The experimental results of section 5.2 are somewhat disappointing. Even with no data augmentation, and even with the original networks, membership can only be assessed with a 90% accuracy. Results are much lower in less favorable cases, sometimes close to random (see last line of Table 3). This seems to be too low to be of practical use. This might be because the Bayes and MAT attacks are too simplistic. Again, why not using the distribution of the outputs of all layers? Why focusing only on the output of the last layer?”
> We agree that better performance could be obtained by running the initial model for more epochs, but our goal is to stay close to the standard training of Imagenet models, i.e. 90 epochs with an initial learning rate of 0.1, divided by 10 every 30 epochs. We emphasize that the last line of Table 3 corresponds to the most difficult setup, where the network has been trained with a strong data-augmentation, and we only use the intermediate layers of the network (which amounts to less than 62% of the parameters for e.g. Resnet101), this is why the performance is significantly impacted. We experimented with more sophisticated models, and it did not bring any improvement (see shadow models in appendix E, e.g. the performance before the softmax layer is 58.2 for Resnet101 and 60.8 for our method).

---

### Official Review · AnonReviewer2 · 2018-11-02
**review of "an empirical evolution of the memorisation properties of Convents"**

**Rating:** 5
**Confidence:** 4

**Review:**

Summary of the paper:


The paper has two intertwined goals. These goals are to illuminate the
generalization/memorization properties of large and deep ConvNets in
tandem with trying to develop procedures related to identifying
whether an input to a trained ConvNet has actually been used to train the
network. The latter task is generalized to detecting if a
particular dataset has been used to train a ConvNet. These goal are
tackled empirically with multiple sets of experiments on largescale
datasets such as ImageNet22k and modern deep ConvNets architectures
such as VGG and ResNet.



Paper's positive points

+ The paper has a very comprehensive set of references in the areas it
touches upon.

+ Some of the experimental results presented are quite
interesting. They show that regularization data-augmentation helps
prevent a network from explicit memorization and could be used as a
way to help make training data more anonymous.

+ Large scale experiments are reported on modern architectures.


Paper's negative points

- The paper makes use of a result from the David MacKay textbook
  which defines the capacity of a single layer network to memorize the
  labelling of $n$ inputs in $d$-dimensional space. If I'm not
  mistaken, from this result the authors extrapolate that the capacity
  of a (deep) neural network is proportional to the number of
  parameters in the network. This is true, but there are a
  couple of caveats. The first is that the coefficient of
  proportionality must depend very much on the number of layers in the
  network. Increasing the network's depth increases the efficiency of
  the representation (i.e. fewer total parameters needed to have the
  same representational power as a shallow network). And as MacKay
  also says in his book (chapter 44 quoting findings from Radford
  Neal) that for MLPs what determines the complexity of the typical
  function (once the network has a large enough width) represented by
  the MLP is the "characteristic magnitude of the weights". So the
  regularization technique applied is very significant in the
  controlling the effective capacity of a network. This paper
  experimentally shows that is the case multiple times as it is shown
  that with increasing degrees of regularization (figure 1, figure 2)
  it becomes harder and harder to memorize the positive training
  images. It would be great if the paper also made some attempt to
  consider these connections. Or at least comment on how these factors
  could be incorporated into a more sophisticated analysis of the
  capacity of a network.



- There is a slight oxymoron in the premise of the first set of
  experiments. The network is forced to memorize a set of
  positive examples relative to the negative set it sees during
  training. What is memorized I presume depends a lot on the negative
  set used for training (its diversity, closeness to the positive set
  and how frequently each negative example is seen during
  training). This issue is not really commented upon in the paper. Is
  there a training task which would allow one to more explicitly
  memorize the image (some sort of reconstruction task) as opposed to
  an in/out classification task?

- This paper is a slightly difficult read - not because of the
  language or the presentation of the material but more because there
  is not one main coherent argument or goal for the paper. This is
  reflected in the "Related work" section where 4 different
  issues/tasks are referred to. Each one of these topics is worthy of
  a paper in itself, but this paper dips into each one and then
  swiftly moves onto the next one. For example in section 3 the paper
  explores if a network can be forced to explicitly memorize a set of
  images and how the size of this set is affected by the number of
  parameters in the network and data augmentation. High-level
  conclusions are made: more parameters in the network implies more
  images can be memorized and data-augmentation makes explicit
  memorization more difficult. Then it is off to considering
  pre-trained networks and determining whether by analyzing the
  statistics of the responses at different layers one can decide if a
  set of images was used for training or not (or similar tasks). Yes
  the different sections are related but it is does not feel like they
  build upon each other to help form a clearer picture of memorization
  within neural networks.


- The conclusions focus on the importance of section 3 and
  the results of the experiments performed. Do the conclusions accurately
  reflect the opinions of the author? If yes, would
  it better to re-organize the paper and devote more of it to the
  material presented in section 3 and filling this out with more
  analysis and experiments to perhaps explore the issue of the
  capacity of a network in more


Queries/ points that need some clarification

- I'm a little unclear when data-augmentation is included in the
  training phase whether the goal is to be able to also recognise
  perturbed versions of the input images at test time. In section 3 is
  a perturbed positive image considered a positive training image? And
  in the testing phase are only unperturbed versions of the positive
  images given to the ConvNet as input?

- Last paragraph page 4: "when the accuracy gets over 60\% and again
  at 90\%". Is this training or validation accuracy?




Typos possible errors spotted along the way:

* First paragraph page 5: "more shallow" --> "shallower"
* Page 7, first paragraph of section 5.: "is ran" --> "is run"
* Using "scenarii" for the plural of "scenario" I would say is pretty
  non-standard and most people would use "scenarios"

---

> ### Author Response · Authors · 2018-11-20
> **Answer to reviewer 2**
>
> We thank the reviewer for their review.
>
> “The paper makes use of a result from the David MacKay textbook which defines the capacity of a single layer network to memorize the labelling of $n$ inputs in $d$-dimensional space. [...] It would be great if the paper also made some attempt to consider these connections. Or at least comment on how these factors could be incorporated into a more sophisticated analysis of the capacity of a network.”
> We agree with the reviewer that our analysis of capacity in section 3 does not take into account the magnitude of the weights, nor the dependence on the depth of the network. Our objective in this section was to provide a empirical lower-bound on the capacity by designing a setup where we can vary the quantity of information contained in a dataset (in our case, N choose n), and evaluate empirically the effect of data augmentation. In relation to section 5, we aim at seeing how much a network can remember if it is explicitly trained to remember a given set of images.
> We understand the limitations of MacKay's analysis, which was presented to give a rough theoretical comparison point to our empirical evaluation. We will clarify this in the paper and improve the discussion along the lines discussed by the reviewer.
>
> “There is a slight oxymoron in the premise of the first set of experiments. The network is forced to memorize a set of positive examples relative to the negative set it sees during training. What is memorized I presume depends a lot on the negative set used for training (its diversity, closeness to the positive set and how frequently each negative example is seen during training). [...] Is there a training task which would allow one to more explicitly memorize the image (some sort of reconstruction task) as opposed to an in/out classification task?”
> In these experiments, the set of positive and negatives is fixed (when varying data augmentation and architectures). During training, we feed to the network all positives and an equal number of negatives during each epoch. The performance does indeed depend on the closeness of the positive and the negatives, but this is similar to the membership inference problem presented in section 5, where it is difficult for a network to tell apart a seen image from an unseen, very similar image. A reconstruction task would suffer the same problems: the reconstruction is only approximate so we would need to evaluate the distance between our reconstruction, positives and negatives, which also depends on the closeness between positives and negatives. Also, the reconstruction task would need to remember the values of all pixels which requires more capacity.
> We agree that this specific deserves a short discussion and will add it to the paper.
>
> “This paper is a slightly difficult read [...] because there is not one main coherent argument or goal for the paper.[...]. Yes the different sections are related but it is does not feel like they build upon each other to help form a clearer picture of memorization within neural networks.”
> The general goal of the paper is to empirically assess memorization in neural networks, and in particular the important question of implicit memorization, which is important for privacy: does a network trained for classification remember an image, or a set of images ? This aspect is empirically evaluated in sections 4 and 5, and section 3 is a preliminary study of the memorization capabilities for systems explicitly trained to memorize (this serves as a qualitative upper-bound for implicit memorization).
> We acknowledge that Section 3, which is a preliminary analysis, may confuse the reader in the first place. We decided to move it to an appendix after reading the feedback from the three reviewers.
>
> “The conclusions focus on the importance of section 3 and the results of the experiments performed. Do the conclusions accurately reflect the opinions of the author?”
> We do not consider the conclusions of the experiments from Section 3  to be more important than those of the other sections, in fact quite the opposite. As mentioned above, we will move it to appendices.
>
> “[...]In section 3 is a perturbed positive image considered a positive training image? And in the testing phase are only unperturbed versions of the positive images given to the ConvNet as input?”
> When data augmentation is used, we consider that perturbed positive images are also positive images.  In the testing phase, perturbed versions of the positive images are given to the ConvNet.
>
> “Last paragraph page 4: "when the accuracy gets over 60\% and at 90\%". Is this training or validation accuracy?”
> We decrease the learning rate when the training accuracy reaches these thresholds.
>
> We thank the reviewer for reporting typos, we will correct them in the paper.

---

### Official Review · AnonReviewer3 · 2018-11-04
**Contributions unclear**

**Rating:** 4
**Confidence:** 2

**Review:**

==============Final Evaluation================
I have gone through the other reviews as well as the author response.
Firstly, I would like to thank the authors for providing detailed responses to my questions.

In general, I agree with R2 that the paper generally has some potentially interesting ideas and results but the manner in which the current draft is organized and presented makes it hard to grasp them and there is a lack of coherent message about what the paper is about.

Moreover, from my understanding the analysis in David McKay’s book (Chapter 41) concerns a single neuron (and the number of parameters for a single neuron). As pointed out by R2, with depth there are a lot more number of possible ways in which one could carve out decision boundaries to separate data points, thus, it is not clear that the loose linear upper bound holds Specifically, as one might expect with depth it could be possible that linear capacity increase is a lower bound (I am not suggesting that it is, but that possibility should be considered and explained in the paper). Similarly, it would be good to formally connect the capacity to the rate of memorization before making a statement about them being related (as suggested in the initial review). In general, I feel this section could use some tighter formalism and justifications.

I also remain unconvinced by the response to my issue with the claim “Our experiments show that our networks can remember a large number of images and distinguish them from unseen images”, where the negative images are also seen by the memorization model, so they are not unseen. The authors address this by saying 3M of the 15 M negatives have been seen. That does not seem like a small enough percentage to claim that these are “unseen” images.

In general, I feel the paper is interesting but would benefit from a major revision which makes the message of the paper more clear, and addresses these and other issues raised in the review phase. Thus I am holding my current rating.
==================

Summary
The paper trains classification models to classify a labeling of a subset of images (assigned with label 1) from the rest of the images (assigned with a label 0). Firstly, the paper shows that deep learning models are able to learn such classifiers and get low training loss. It then proposes to use this model to ``attack’’ task-specific models to perform membership inference, i.e. figuring out if an image provided in a set was used in training or not.

Strengths
+ The paper thoroughly covers related work and provides context.
+ Results on confidence as a signature of a dataset are interesting.

Weaknesses

[Motivation]
1. In general, recent work has found that the raw number of parameters has little to do with the size of the model class or the capacity of a model for deep models, and thus work like [A] has been trying to come up with better complexity measures for models to explain generalization. Thus, without sufficient justification the assertion in the paper that the capacity of the network is well approximated by the number of parameters does not seem correct. Also, the claim in Fig. 1 that the transition from ‘’high capacity’’ to low capacity happens at the number of parameters in the network seems a bit loose and hard to substantiate from what I understand, and should be toned down. (*)

[Capacity]
2. Sec. 3.3, Fig. 3: The capacity (in terms of parameters)of both Resnet-18 and VGG-16 is higher than the capcity for YFCC100M dataset for n=10K images (comes to 161K bits), while the capacity of Resnet-18, with 14.7 million parameters (assuming float32 encoding) has 14.7 * 32 bits = 470.4 million bits, thus capacity alone cannot explain why VGG converges faster than Resnet-18, since both networks exceed the capacity, and capacity does not seem to have an established formal connection to rate of memorization. This is something which would need to be explained/ substantiated separately. (*)

3. Scenario discussed in Sec. 4 seems somewhat impractical. Given a set of m images, it is not clear that a classifier that is trained to detect between train and validation is sufficient, as one might also need to figure out if it is neither train nor val, which is a very practical scenario.

4. Fig. 3 (right): It is not clear why the fact that the classifier is able to predict which dataset the image ‘m’ corresponds to is useful or practical, as this seems to be a property of the set ‘m’ rather than the property of the trained classification model (f_\theta). Please clarify. On the other hand it is clear that using the confidence of the model to predict the dataset is a useful property, but the right side of the Fig. is very confusing. (*)

6. It is not clear to me what the point of Sec. 5 is, given a trained model, one wants to figure out if an image was present in the training of the model. While the baseline approaches seem to make use of the model confidence, I cannot see how the proposed approach (which uses a classifier) makes use of the original model. It is also not clear why Table. 3 does not report the Bayes baseline results. Also, does this section use the classifier for predicting the dataset, or is the approach reported in the section, the MAT approach?

7. ``Our experiments show that our networks can remember a large number of images and distinguish them from unseen images’’ -- this does not seem to be true, since the model is trained on both n as well as N -n ``unseen’’ images which it labels as the negative class, thus the negative class is also seen by the memorization model. (*)

Minor Points
1. It is not clear that training a network to classify a set from another set is necessarily equivalent to ``memorization’’. In addition, the paper would also need to show that such a model does not generalize to a validation set of images. This is probably obvious given the results from Zhang et.al. but should be included as a sanity check.
2. Figure 3: it is confusing to call the cumulative distribution of the maximum classification score as the CDF of the model (y-axis fig. 3 left) as CDF means something else generally in such contexts, as the CDF of a predictor.


References:
[A]: Blier, Léonard, and Yann Ollivier. 2018. ``The Description Length of Deep Learning Models.’’ arXiv [cs.LG]. arXiv. http://arxiv.org/abs/1802.07044.

Preliminary Evaluation
There are numerous issues with the writing and clarity of the paper, while it seems like some of the observations around the confidence of classifiers are interesting, in general the connection between those set of results and the ``memorization’’ capabilities of the classifier trained to remember train vs val images is not clear in general. Important points for the rebuttal are marked with (*).

---

> ### Author Response · Authors · 2018-11-20
> **Answer to reviewer 3**
>
> We thank the reviewer for the detailed comments.
>
> “1.[...] without sufficient justification the assertion in the paper that the capacity of the network is well approximated by the number of parameters does not seem correct. Also, the claim in Fig. 1 that the transition from ‘’high capacity’’ to low capacity happens at the number of parameters in the network seems a bit loose and hard to substantiate from what I understand (*)”
> We agree raw parameter count is not a fine estimate of the capacity of the network. However, an information-theoretic argument shows that an upper-bound of the capacity is the raw parameter count times the size of the representation (i.e. 32 bits for float32, this argument is close to that of [A]). Experimentally, we show that networks with no data-augmentation (figure 1 - purple curve) stop fitting perfectly when the parameters get within 1/10 of the quantity of information in the learning set, thus we think that raw parameter count is a good first-order approximation up to that factor.
>
> “2. Sec. 3.3, [...] capacity alone cannot explain why VGG converges faster than Resnet-18 [...]”
> We observe that the rate of memorization depends on the architecture and the optimization algorithm, but predicting or explaining this rate is beyond the scope of this paper.
>
> “(*) 3. Scenario discussed in Sec. 4 seems somewhat impractical. [...] one might also need to figure out if it is neither train nor val”
> In section 4, we do not train a classifier to distinguish between a training and a validation set. Rather, we use a readily-available classifier (trained for e.g. image recognition) for a completely different purpose than what it was trained for, i.e. to distinguish datasets of images (section 4.1) or detect if a set of images comes from a given set (section 4.2). Section 4.2 shows how to use the K-S test to detect leakage, but the same test could tell if the m-set comes from neither the train nor the validation sets.
>
> “4. Fig. 3 (right): It is not clear why the fact that the classifier is able to predict which dataset the image ‘m’ corresponds to is useful or practical, as this seems to be a property of the set ‘m’ rather than the property of the trained classification model (f_\theta). Please clarify. [...]”
> Being able to tell from the classifier output (using e.g. the confidence) if a set of images comes from the training or the validation set is a good indicator of how much the network has memorized these images. In our opinion, the most important outcome of the experiment of section 4.1 (figure 3, right) is to determine how many samples are needed to reliably discriminate the training set from the validation set (this corresponds to the solid curves), which is related to how much the model has memorized images from the training set.
>
> “6. Does [section 5] use the classifier for predicting the dataset, or is the approach reported in the section, the MAT approach?”
> The baseline approaches make use of the loss of the model (which is not the same as the confidence). The proposed approach uses the lower layers of the original model, and upper layers learnt on a separate, public set (this is the “partial-layers” setting). Table 3 reports the results of the Bayes method on top of this network with upper layers retrained, as the MAT usually gives similar results on this task (for instance, Table 4 reports 60.8% performance with Softmax + Flip, Crop on Resnet101 for the Bayes method, and the MAT gets 61.14%).
>
> “7. ``Our experiments show that our networks can remember a large number of images and distinguish them from unseen images’’ -- this does not seem to be true, since the model is trained on both n as well as N -n ``unseen’’ images which it labels as the negative class, thus the negative class is also seen by the memorization model. (*)”
> We feed our model an equal number of positives and negatives (chosen randomly) at each epoch. For n < 10K, after 300 epochs the model has seen at most 3M negatives out of 15M, and yet still generalizes to the unseen negatives.
>
> “1. It is not clear that training a network to classify a set from another set is necessarily equivalent to ``memorization’’. In addition, the paper would also need to show that such a model does not generalize to a validation set of images. [...]”
> With the downstream application of sections 4 and 5, we are interested in “memorization” in the sense of any classifier that can tell apart images marked as “positives” from images marked as “negatives”. This notion is somewhat different from
> memorization as defined in other papers, where it is related to having a good training accuracy and a a validation accuracy close to random guessing. With the setup used in section 3, there is no good notion of validation: our model is expected to predict “0” on held-out data.
>
> “2. Figure 3: [the term CDF] is confusing”
> We will update the caption to make it less ambiguous.

---

### Author Response · Authors · 2018-11-26
**Update of the paper**

We have updated the paper to include changes suggested by the reviewers: we modified section 3 to include a general framework that encompasses sections 4 and 5, and reduce the part devoted to explicit memorization.

---

### Meta-Review · Area_Chair1 · 2018-12-15

**Confidence:** 5
**Recommendation:** Reject

**Metareview:**

This paper studies memorization properties of convnets by testing their ability to determine if an image/set of images was used during training or not. The experiments are reported on large-scale datasets using high-capacity networks.

While acknowledging that the proposed model is potentially useful, the reviewers raised several important concerns that were viewed by AC as critical issues:
(1) more formal justifications are required to assess the scope and significance of this work contributions -- see very detailed comments by R2 about measuring networks capacity to memorize and the role of network weights and depth as studied in MacKay,2002. In their response the authors acknowledged they didn’t take into account network weights and depth but strived at an empirical evaluation scenario.
(2) writing and presentation clarity of the paper could be substantially improved – see very detailed comments by R3 and also R2;
(3) empirical evaluations and effect of the negative set used for training are not well explained and analysed (R2, R3).

AC can confirm that all three reviewers have read the author responses and have contributed to the final discussion.
AC suggests, in its current state the manuscript is not ready for a publication. We hope the reviews are useful for improving and revising the paper.